# LEARNING NEURAL CAUSAL MODELS FROM UNKNOWN INTERVENTIONS

## ABSTRACT

Meta-learning over a set of distributions can be interpreted as learning different types of parameters corresponding to short-term vs long-term aspects of the mechanisms underlying the generation of data. These are respectively captured by quickly-changing *parameters* and slowly-changing *meta-parameters*. We present a new framework for meta-learning causal models where the relationship between each variable and its parents is modeled by a neural network, modulated by structural meta-parameters which capture the overall topology of a directed graphical model. Our approach avoids a discrete search over models in favour of a continuous optimization procedure. We study a setting where interventional distributions are induced as a result of a random intervention on a single unknown variable of an unknown ground truth causal model, and the observations arising after such an intervention constitute one meta-example. To disentangle the slow-changing aspects of each conditional from the fast-changing adaptations to each intervention, we parametrize the neural network into fast parameters and slow meta-parameters. We introduce a meta-learning objective that favours solutions *robust* to frequent but sparse interventional distribution change, and which generalize well to previously unseen interventions. Optimizing this objective is shown experimentally to recover the structure of the causal graph. Finally, we find that when the learner is unaware of the intervention variable, it is able to infer that information, improving results further and focusing the parameter and meta-parameter updates where needed.

## 1 INTRODUCTION

A major challenge of contemporary deep learning is to generalize well outside the assumptions of independent and identically distributed data, when we care about generalization or fast adaptation to distributions other than the main training distribution. For this purpose, we propose an approach that starts by distinguishing between: (a) an underlying causal model, (b) *observational* distributions derived from it, an (c) *interventional* distributions arising from *interventions* upon its variables, whether they be known or unknown. Fortunately, these distinctions can be addressed by the paradigm of Structural Causal Models (SCMs) (Pearl, 1995; Peters et al., 2017) and a wide body of associated literature. Unlike other frameworks using SCMs we also account for interventions performed by agents other than an experimenter. By treating the interventions of other agents as unknown interventions that lead to changes in the underlying data distribution, the present work is a contribution towards the use of a meta-learning for causal model induction.

Estimating the underlying causal structure from data is an open and challenging problem (Pearl, 2009; Imbens & Rubin, 2015). A lot of prior work has examined learning causal structure based on observational data (Chickering, 2002; Tsamardinos et al., 2006; Goudet et al., 2017; Hauser & Bühlmann, 2012; Spirtes et al., 2000; Sun et al., 2007; Zhang et al., 2012; Shimizu et al., 2006; Hoyer et al., 2009; Daniusis et al., 2012; Budhathoki & Vreeken, 2017; Kalainathan et al., 2018). However, many real-world datasets have an inherent distributional heterogeneity due to different interventions to the variables composing the model. In these situations interventional approaches are needed (e.g. Heckerman et al., 1995; Cooper & Yoo, 1999; Hauser & Bühlmann, 2012; Peters et al., 2016; Rothenhäusler et al., 2015; Ghassami et al., 2017). Established approaches for causal inference are often either relying on restrictive assumptions or on conditional independence testing, which is hard (Shah & Peters, 2018). Furthermore, most of these approaches either assume full knowledge of the intervention or make strong assumptions about its form (Heinze-Deml et al., 2018).

However, in the real world, interventions are also not always performed by an experimenter. They can be performed by other agents, or by environmental changes in ways that are unknown, or by a naive learner (like a robot) which does not know precisely yet how its low-level actions change high-level causal variables. In this paper, we look at the setting where interventions are unknown, and our goal is to discover causal graphs given unknown-intervention samples. The challenging aspect of this setting is to not only learn the causal graph structure, but also predict the intervention accurately. In this setting, we need to make sure to: (1) avoid an exponential search over all possible DAGs, (2) handle unknown interventions, (3) model the effect of interventions, and (4) model the underlying causal structure.

One possibility for learning a causal structure (through SCM modelling) is to perform many *experiments* in which one executes interventions. Thus, such interventions modify the effect of the intervened upon variable from its parents in the corresponding DAG, which the model has to quickly adapt to. We can make parallel connections to meta-learning, where the inner loop can be considered as fast adaptation to the distribution change, and outer loop can be considered as learning the stationary meta-parameters of the model. For causal induction, one can consider each distribution which arises as a result of an intervention as a meta-example, and use a meta-learning objective for fast adaptation in response to an intervention. One can think of model parameters as being composed of slow- and fast-changing parameters. The slow parameters are analogous to the meta-parameters in meta-learning and are used for (1) intervention prediction in order to handle the unknown intervention and for (2) modeling the underlying causal structure. On the other hand, the fast parameters are used to model the effect of interventions. An explicit search over the exponentially-growing space of all possible DAGs is avoided by modeling the conditionals of the structural causal model using function approximators, with one neural network per variable. The belief over whether one node $i$ is a direct causal parent of another node $j$ corresponds to a dropout probability for the $i$-th input of network $j$ (which predicts variable $j$ given its parents). This cheaply represents all $2^{M^2}$ possible model graphs, with the graph search implicitly achieved by learning these dropout probabilities. We thus propose a new method for fast adaptation and learning of neural causal models by framing the problem in a meta-learning setting, similar to (Dasgupta et al., 2019; Bengio et al., 2019).

**Our contributions** Our key contributions can be summarized as follows:

- Handle causal induction to the case where the variable on which a soft intervention took place is not known by the learner, and show that better results can be obtained when the learner attempts to infer that information and uses it to appropriately change parameters and meta-parameters.

- We bypass the issue of having to optimize over and represent an exponentially large set of discrete causal graphs by learning an efficiently parametrized ensemble of SCMs,

- Show that our algorithm correctly identifies the causal graph and use the learned graph for generalization to an unseen environment.

## 2 PRELIMINARIES

A Structural Causal Model (SCM) (Peters et al., 2017) over a finite number $M$ of random variables $X_i$ is a set of structural assignments

$$X_i := f_i(X_{pa(i,C)}, N_i), \quad \forall i \in \{0, \dots, M-1\} \tag{1}$$

where $N_i$ is jointly-independent noise and $pa(i, C)$ is the set of parents (direct causes) of variable $i$ under configuration $C$ of the SCM directed acyclic graph, i.e., $C \in \{0, 1\}^{M \times M}$, with $c_{ij} = 1$ if node $i$ has node $j$ as a parent (equivalently, $X_j \in X_{pa(i,C)}$; i.e. $X_j$ is a direct cause of $X_i$). The $n$-th power of an adjacency matrix, $C^n$, counts the number of length-$n$ walks from node $i$ to node $j$ of the graph in element $c_{ij}$. The trace of the $n$-th power of an adjacency matrix, $\text{Tr}(C^n)$, counts the number of length-$n$ cycles in the graph. Causal structure learning is the recovery of the ground-truth $C$ from observational and interventional studies.

**Functional and structural meta-parameters** Let us consider the simplest SCMs, those with $M = 2$ random variables (lets say random variables $A$ and $B$). Only three DAGs exist relating them; they are $A \rightarrow B$, $B \rightarrow A$ or $A \perp B$. These can be represented as the following. If $A$ causes

$B$, then the SCM between $A$ and $B$ can be represented as $B = f_{\theta_B}(c_{BA} \cdot A, \epsilon_B)$, where $c_{BA} = 1$, $\epsilon_B \sim N_B$ is a sample of an independent noise factor. If $B$ causes $A$, then the SCM can be presented as $A = f_{\theta_A}(c_{AB} \cdot B, \epsilon_A)$, where $c_{AB} = 1$, $\epsilon_A \sim N_A$ is a sample of an independent noise factor. If $A$ and $B$ are independent, then we have the same equations but we set $c_{AB} = 0$ and $c_{BA} = 0$.

We may now think of learning the structural causal model as learning the two probabilities of $c_{AB}$ or $c_{BA}$ being 1 (versus 0), representing our belief in the causal relationship between $A$ and $B$. We can parametrize these probabilities differentiably using $P(c_{AB} = 1) = \sigma(\gamma_{AB})$ and $P(c_{BA} = 1) = \sigma(\gamma_{BA})$, with $\gamma$ real numbers and $\sigma(x) = \frac{1}{1+e^{-x}}$. We can also parametrize the structural equations $f_{\theta_A}$ and $f_{\theta_B}$ in a differentiable manner, using conditional probability tables (CPTs) or neural networks.

Hence, the problem becomes simultaneously learning the *structural* meta-parameters $\gamma$ and the *functional* meta-parameters $\theta$. Functional meta-parameters $\theta$ may be easily learned by maximum likelihood or a proxy, and using backpropagation. However, the $\gamma$ are more difficult to infer. Bengio et al. (2019) propose to learn them by using observations of how the distribution changes sparsely, with the transfer generalization (the adaptation rate after the sparse change) being the training objective for $\gamma$. Simultaneously inferring both $\theta$ and $\gamma$ is still more difficult. An $M$-variable SCM over random variables $X_i$, $i \in \{0, \ldots, M-1\}$ can induce a super-exponential number of adjacency matrices $C$. The super-exponential explosion in the number of potential graph connectivity patterns and the super-exponentially growing storage requirements of their defining conditional probability tables make CPT-based parametrizations of the structural assignments $f_i$ increasingly unwieldy as $M$ scales. As shown below, neural networks with $c_{ij}$-masked inputs can provide a more manageable parametrization. For more background about different kinds of intervention we ask the reader to refer Appendix A.3.

## 3 PROPOSED FRAMEWORK: META LEARNING FOR CAUSAL INDUCTION

Our framework disentangles the slow-changing meta-parameters, which reflect the stationary properties discovered by the learner, and the fast-changing parameters, which adapt in response to interventional changes in distribution. We consider two kinds of meta-parameters: the causal graph structure $\gamma$ and the model's slow weights $\theta_{\text{slow}}$, along with the meta-learning objective for both of them. We also consider one kind of parameter: the model's fast weights, $\theta_{\text{fast}}$. We will call $\theta = \theta_{\text{slow}} + \theta_{\text{fast}}$ the sum of the slow, stationary meta-parameters and the fast, adaptational parameters.

### 3.1 TASK DESCRIPTION

Our task setup deviates from most common deep learning modeling setups. For the purposes of this work, we restrict ourselves to inference of randomly-generated or manually-provided ground-truth SCMs of $M$ categorical random variables causally related via a DAG. The model is permitted to see (1) data from the original ground-truth model, and (2) data from a modified ground-truth model with a random intervention applied. In our experiments, at most one intervention is concurrently performed. When an intervention is performed, a single node is randomly and uniformly chosen among all $M$ nodes, and its ground-truth distribution soft-intervened upon. The learner model is aware of the samples having come from an intervention distribution, but is not aware of the identity of the intervention node, and so must predict it. Each run of sampling steps under a given intervention is referred to as an *episode*. The learner, over a large number of episodes, will experience all nodes being intervened upon, and should be able to infer the SCM from these interventions.

### 3.2 CAUSAL INDUCTION AS AN OPTIMIZATION PROBLEM

We first explain how we mitigate the problem of searching in the super-exponential set of graph structures. If there are $M$ such variables, the strategy of considering all the possible structural graphs as separate hypotheses is not feasible because it would require maintaining $O(2^{M^2})$ models of the data. We note that we can cheaply choose any of the $2^{M^2}$ possible DAGs through suitable independent Bernoulli choices $c_{ij}$ associated with each edge $i \to j$ of the causal graph, i.e., sampling all the $c_{ij}$'s independently. Then we only need to learn the $M^2$ coefficients $\gamma_{ij}$, and we implicitly maintain a distribution over the $2^{M^2}$ models corresponding to all the possible draws of $c_{ij}$. Note that a slight dependency between the $c_{ij}$ is induced if we require the causal graph to be acyclic (which

allows one to sample the $X$ using ancestral sampling). To enforce that constraint it is not sufficient to require $c_{ij}c_{ji} = 0$ (both cannot be 1). We deal with this problem with a regularizer acting on the $\gamma$'s in order to favour acyclic solutions (Zheng et al., 2018).

In our approach, each random variable's structural assignment is modeled via $X_i := f_{\theta_i}(c_{i0} \times X_0, c_{i1} \times X_1, ..., c_{im} \times X_m, \epsilon_i)$, where $f_{\theta_i}()$ is a neural network (MLP) with parameters $\theta_i$, and $c_{ij} \sim \text{Bin}(\text{sigmoid}(\gamma_{ij}))$. Through this construction we can frame the causal induction problem as an optimization problem, with $\theta$ optimized to maximize the likelihood of data under the model but $\gamma$ optimized with respect to a meta-learning objective arising from changes in distribution because of interventions. There are a few benefits for learning a parametrized ensemble of SCMs. Such an ensemble is analogous to an ensemble of neural nets differing by their binary input dropout masks, which select what variables are used as predictors of another variable.

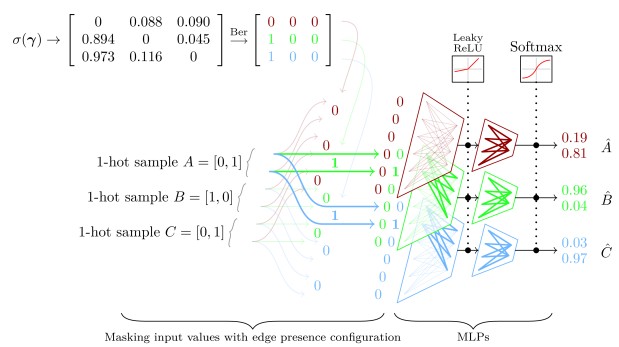

Figure 1: MLP Model Architecture for $M = 3$, $N = 2$ (`fork3`) SCM. The model computes the conditional probabilities of $\hat{A}, \hat{B}, \hat{C}$ given their parents using a stack of three independent MLPs. The MLP input layer uses an adjacency matrix sampled from $\text{Ber}(\sigma(\gamma))$ as an input mask to force the model to make use only of parent nodes to predict their child node.

### 3.3 Fast Adaptation by Meta-Learning

**Fast and slow weights** To disentangle an environment's stable, unchanging properties (the causal structure) from unstable, changing properties (the effects of an intervention), we proposed in §3 to distinguish between the model's functional meta-parameters $\theta_{\text{slow}}$ and parameters $\theta_{\text{fast}}$. The sum of these weights, $\theta = \theta_{\text{slow}} + \theta_{\text{fast}}$, parametrizes the MLPs computing the conditionals $P_i(X_i|X_{pa(i)}; \theta_i)$. The fast weights and the slow weights terminology is drawn from Hinton & Plaut (1987). The construction of $\theta$ as a sum of initial, slow weights plus zeroed, fast weight that are then allowed to quickly adapt during a transfer episode is due to MAML (Finn et al., 2017). The ability to generalize out-of-distribution by adapting to a transfer distribution can then be measured by the likelihood after adapting the fast weights on transfer data.

Since an intervention is generally not persistent from one transfer distribution to another, the model's functional parameters ($\theta_{\text{fast}}$) are reset after each episode of transfer distribution adaptation. The meta-parameters ($\theta_{\text{slow}}, \gamma$) are preserved, then updated after each episode. Inspired by Bengio et al. (2019), the meta-objective for each meta-example over some intervention distribution $D_{\text{int}}$ is the following[1] "meta-transfer" loss:

$$\mathcal{R} = -\mathbb{E}_{X \sim D_{\text{int}}}[\log \mathbb{E}_{C \sim \text{Ber}(\gamma)}[\prod_i \mathcal{L}_{C,i}(X; \theta_{\text{slow}})]] \qquad (2)$$

where $X$ is an example sampled from the intervention distribution $D_{\text{int}}$, $C$ is an adjacency matrix drawn from our belief distribution (parametrized by $\gamma$) about graph structure configurations and

$$\mathcal{L}_{C,i}(X) = P(X_i|X_{pa(i,C)}; \theta_{\text{slow}}) \qquad (3)$$

is the likelihood of the $i$-th variable $X_i$ of the sample $X$, when predicting it under the configuration $C$ from the set of its putative parents, $X_{pa(i,C)}$.

**Structural Parameter Gradient Estimator** Because a discrete Bernoulli random sampling process is used to produce the configurations under which the log-likelihood of data samples is obtained, we require a gradient estimator to propagate gradient through to the $\gamma$ structural meta-parameters.

---

[1] These equations differ from Bengio et al. (2019) in that the $t$ indices and products were dropped, because the computation is not online.

We adopt for this purpose the gradient estimate of Bengio et al. (2019) *(but see footnote 1)*:

$$g_{ij} = \frac{\sum_k (\sigma(\gamma_{ij}) - c_{ij}^{(k)}) \mathcal{L}_{C,i}^{(k)}(X)}{\sum_k \mathcal{L}_{C,i}^{(k)}(X)} \tag{4}$$

where the $^{(k)}$ superscript indicates the values obtained for the $k$-th draw of $C$. This gradient is estimated solely with $\theta_{\text{slow}}$ because estimates employing $\theta$ have much greater variance.

**Acyclic Constraint** We add a regularization term to the loss term that discourages the model from having length-2 cycles in the learned adjacency matrix. The regularizer term is

$$J_{\text{DAG}} = \sum_{i \neq j} \cosh(\sigma(\gamma_{ij})\sigma(\gamma_{ji})), \quad \forall i, j \in \{0, \dots, M-1\}$$

and is derived from Zheng et al. (2018). The details of the derivation are in the Appendix.

### 3.4 MODEL DESCRIPTION

**Learner Model** We model the $M$ structural assignments $X_i := f_i(X_{pa(i,C)}, N_i)$ (Eq. 1) of the SCM as a set of $M$ multi-layer perceptrons (MLPs), as in Bengio et al. (2019). The MLPs are identical in shape but do not share any parameters, since they are modeling *independent causal mechanisms*. Each possesses an input layer of $M \times N$ neurons (for $M$ one-hot vectors of length $N$ each), a single hidden layer chosen arbitrarily to have $\max(4M, 4N)$ neurons with a LeakyReLU activation of slope 0.1, and an output layer of $N$ neurons representing the unnormalized log-probabilities of each category. To force $f_i$ to rely exclusively on the direct ancestor set $pa(i, C)$ under adjacency matrix $C$ (See Eqn. 2), the one-hot input vector $X_j$ for variable $X_i$'s MLP is masked by the Boolean element $c_{ij}$. The functional parameters of the MLP are the set $\theta = \{\texttt{W0}_{ihjn}, \texttt{B0}_{ih}, \texttt{W1}_{inh}, \texttt{B1}_{in}\}$.

An example of the multi-MLP architecture with $M = 3$ categorical variables of $N = 2$ categories is shown in Figure 1.

**Ground-Truth Model** In our experiments, ground-truth SCM models are parametrized as a set of MLPs of the same size as the learner models, thus avoid having to manually define the Conditional Probability Tables (CPTs). Ground-truth models exist in two variants that differ mainly in initialization: *synthetic*, where the $\theta$ are randomly initialized and the $\gamma$ are either randomly-initialized or pre-specified; and *real-world*, where the $\theta$ and $\gamma$ are both initialized so as to closely replicate the CPTs of given Bayesian networks.

### 3.5 INTERVENTIONS

**Soft interventions** To execute an intervention on variable $X_i$, we reinitialize $X_i$'s ground-truth MLP parameters randomly while leaving other variables' MLPs untouched. A copy of the old parameters is saved, allowing the intervention to be undone by resetting the parameters back to their original values.

**Predicting interventions** After an intervention on $X_i$, the gradients into the learned model's $\gamma_i$ and the slow weights for the $i$-th conditional are false, because they do not bear the blame for $X_i$'s outcome (which lies with the intervener). We find that ignoring this issue considerably hurts or slows down meta-learning, suggesting that we should try to infer on which variable the intervention took place. For this purpose, we take advantage of the fact that the conditional likelihood of the intervened variable tends to have a poorer relative likelihood under $D_{\text{int}}$, so we pick the variable with the greatest deterioration in likelihood as our guess.

### 3.6 TRAINING ALGORITHM

The structural meta-parameters $\sigma(\gamma_{ij})$ represent the belief in the hypothesis that node $i$ has node $j$ as a direct causal parent. We may sample from this belief, obtaining different configurations (causal structures) of the causal graph. We hypothesize that the correct configuration enables better adaptation to a slight change in distribution, e.g. resulting from a soft intervention. Hence, we evaluate different configurations under the transfer distribution; those giving a higher transfer likelihood under $\theta_{\text{slow}}$ get

a higher reward and their probability is increased. The functional (meta-)parameters are trained as usual by gradient ascent on the log-likelihood. The details of the training algorithm is in Section A.2 in the Appendix.

Synthetic datasets use either a specified edge structure for $\gamma$, or randomly initialize $\gamma_{ij}$ such that it is Boolean, strictly lower-triangular, and each row has an expected sum of 1-5 (and therefore each node expects 1-5 direct ancestors). Real-world datasets, specified as CPTs, must first be converted or approximated by a near-ground-truth MLP. We use the graph's proper edge structure to initialize $\gamma$, and learn $\theta$ by training individually each MLP in the set to replicate the correct pre-softmax logits. In practice, excellent reproductions of the CPT can be achieved.

**Stability of training**   Our model requires simultaneous training of both the structural and the functional meta-parameters, but these are not independent and do influence each other, which leads to instability in training. For example, if $\sigma(\gamma_{ij}) \approx 0$ incorrectly, the $i$-th MLP does not learn to use input $X_j$, and vice-versa, if the $i$-th MLP has not learned to properly use input $X_j$, this will favour pushing $\sigma(\gamma_{ij})$ towards 0. To overcome this instability, we pretrain the model under observational data (from the distribution of the data before interventions) using dropout on the inputs. This ensures that the functional meta-parameters $\theta_{\text{slow}}$ are not too biased towards certain configurations of the meta-parameters $\gamma$.

## 4    RELATED WORK

The recovery of the underlying structural causal graph from observational and interventional data is a fundamental problem (Pearl, 1995; 2009). Different approaches have been studied, score-based, constraint-based and asymmetry-based methods. Score-based methods search through the space of all possible directed acyclic graphs (DAGs) representing the causal structure based on some form of scoring function for network structures (Chickering, 2002; Tsamardinos et al., 2006; Goudet et al., 2017; Hauser & Bühlmann, 2012; Heckerman et al., 1995; Cooper & Yoo, 1999; Hauser & Bühlmann, 2012). Constraint-based methods (Spirtes et al., 2000; Sun et al., 2007; Zhang et al., 2012) infer the DAG by analyzing the conditional independence of data. Eaton & Murphy (2007b) use dynamic programming techniques to accelerate Markov Chain Monte Carlo (MCMC) sampling in a Bayesian approach to structure learning for discrete variable DAGs. Asymmetry-based methods (Shimizu et al., 2006; Hoyer et al., 2009; Daniusis et al., 2012; Budhathoki & Vreeken, 2017; Mitrovic et al., 2018) assume asymmetry between cause and effect in the data and try to use this information to estimate the causal structure. Recently Peters et al. (2016); Ghassami et al. (2017) proposed to exploit invariance across different environments to infer causal structure, but are difficult to scale to large graphs due to the necessary iteration over the super-exponential set of possible graphs.

For interventional data, it is often assumed that the models have access to full intervention information, which is rare in the real world. Rothenhäusler et al. (2015) have investigated the case of additive shift interventions, while Eaton & Murphy (2007a) have examined the situation where the targets of experimental interventions are imperfect or uncertain. This is different from our setting where the intervention is unknown to start with and is assumed to arise from other agents and the environment.

Learning based methods have been proposed (Guyon, a;b; Lopez-Paz et al., 2015) and there also exist recent approaches using the generalization ability of neural networks to learn causal signals from purely observational data (Kalainathan et al., 2018; Goudet et al., 2018). Neural network methods equipped with learned masks, such as (Ivanov et al., 2018; Li et al., 2019; Yoon et al., 2018; Douglas et al., 2017), exist in the literature, but only a few (Kalainathan et al., 2018) have been adapted to causal inference. This last work is, however, tailored for causal inference on continuous variables and from observations only. Adapting it to a discrete-variable setting is made difficult by its use of a Generative Adversarial Network (GAN) Goodfellow et al. (2014) framework.

Turning now to meta-learning, Dasgupta et al. (2019) have used it to learn to make predictions under interventions. However, their approach does not induce a causal graph, neither explicitly nor via decoding. Thus, it cannot be used for general causal discovery, but only to make predictions of variable values. Most similar to our work, Bengio et al. (2019) proposes a meta-learning framework for learning causal models from interventional data. However, the proposed method (Bengio et al., 2019) explicitly models every possible set of parents for every child variable and attempts to distinguish the best among them. Because there are combinatorially-many such parent sets, the method cannot scale beyond trivial graphs. In our work, we bypass this restriction by modeling the edge between any 2 variables as a dropout probability and hence our model only scales quadratically with the graph size.

## 5 EXPERIMENTAL SETUP AND RESULTS

Our experiments aim to evaluate the proposed method to recover the correct causal structure and which elements of the method matter. We first evaluate our model on a synthetic dataset where we have control over the number of variables and causal edges in the ground-truth SCM. This allows us to verify after learning with what accuracy we recover the individual decisions about $c_{ij}$ and understand the performance of our algorithm under various conditions. We then evaluate our method on real world datasets collected from the BnLearn dataset repository, and show that the proposed approach recovers the true causal structure. We then show that the trained models can correctly predict the consequences of previously unseen interventions on the rest of the graph. We also perform ablations showing how important is each component of the model.

### 5.1 SYNTHETIC DATASETS

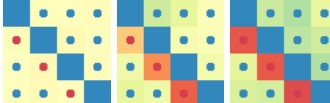 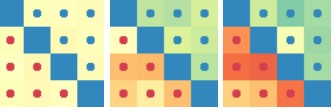

Figure 2: Learned edges at three different stages of training. **Left**: Chain graph with 4 variables. **Right**: Fully-connected DAG graph with 4 variables.

We first evaluate the model's performance on several randomly-initialized SCMs with specific, representative graph structures. For $M = 3$-variable DAGs, we consider every possible connected graph: `chain3`, `fork3`, `collider3` and `confounder3` (See Fig. 7 in appendix). They exhibit every graph sub-structure that can exist in larger graphs, and must be mastered before tackling larger graphs. Since the number of possible DAGs grows super-exponentially with the number of variables, for $M > 3$ up to $8$ a selection of representative and edge-case graphs are chosen. The `chainM` and `fullM` graphs ($M = 3\text{-}8$) are the minimally and maximally connected M-variable graphs, while the remaining graphs are randomly generated with a varying sparsity level (1-4 expected number of parents per node). The details of the setup can be found in Appendix A.5.

**Results** The model can successfully recover the correct edge structure for all synthetic graphs considered. The learning curves plotting the average cross-entropy (CE) loss for the learned edges against the ground-truth model for $M = 3$ are shown in Figure 7. The fully-connected `confounder3` graph is particularly easy to learn, but all 3-variable graphs are learned perfectly. Plots of the same process on $4$- through $8$-variable graphs were similarly encouraging, with all models converging to a negligible loss. The results are, however, sensitive to some hyperparameters, notably the DAG penalty and the sparsity penalty.

### 5.2 REAL-WORLD DATASETS: BNLEARN

The Bayesian Network Repository is a collection of commonly-used causal Bayesian networks from the literature, suitable for Bayesian and causal learning benchmarks. We evaluate our model on the Earthquake (Korb & Nicholson, 2010), Cancer (Korb & Nicholson, 2010) and Asia (Lauritzen & Spiegelhalter, 1988) datasets ($M = 5$, 5 and 8-variables respectively, maximum 2 parents per node) in the BnLearn dataset repository.

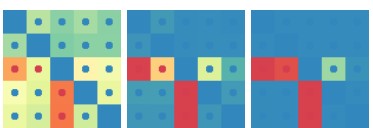

Figure 3: Earthquake: Learned edges at three different stages of training.

The ground-truth SCM is given for each dataset, and the functional parameters are represented as conditional probability tables (CPTs). We learn a near-ground-truth MLP from the dataset's CPT and use it as the ground-truth data generator. We also insert a (greater than 1) temperature factor in order to increase the likelihood of sampling some very rare events in the CPTs. Details of the setup can be found in Appendix A.5.1.

**Results** The model can successfully recover the correct edge structure for all BnLearn graphs considered up to and including 8-variable Asia. Figures 3 and 4 illustrate what the model has learned at several stages of learning. In these figures, the $\sigma(\gamma_{ij})$ and $c_{ij}$ adjacency matrix elements are plotted as colored squares and dots. Off-diagonal terms are unknown, and appear yellow.

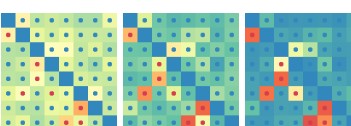

Figure 4: Asia: Learned edges at three different stages of training.

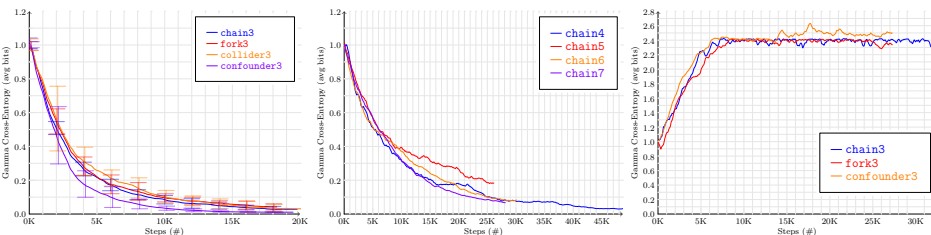

Figure 5: **Left**: Cross entropy (CE) for edge probability between learned and ground-truth graphs for all 3-variable SCMs. Error bars are $\pm 1\sigma$ over PRNG seeds 1-5. **Middle**: Edge CE loss for the chain graph with 4-7 variables. **Right**: Edge CE loss for 3-variable graphs with no dropout during pretraining, showing the importance of this dropout.

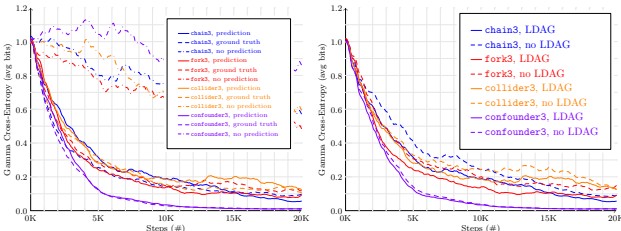

Figure 6: Ablations study results on all possible 3 variable graphs. Both graphs show the cross-entropy loss on learned vs ground-truth edges over training time. **Left**: Models that infer the intervention (prediction, bold) vs models that have knowledge of the true intervention (ground truth, long dash) vs models that use no knowledge of the intervention at all (no prediction, short dash). Result suggests inferring the intervention works almost as well as knowing the true intervention. **Right**: Comparisons of model trained with and without DAG regularizer ($L_{\text{DAG}}$), showing that DAG regularizer helps convergence.

The color of the squares indicates belief in the presence or absence of an edge ($\sigma(\gamma_{ij})$), while the color of the dot indicates the ground truth $c_{ij}$. Red indicates (belief in) an edge; Blue indicates (belief in) the absence of an edge. Yellow indicates maximum uncertainty At the beginning of training, the main diagonal is *a priori* known to be clear of edges, and therefore is solid blue.

As training progresses, the beliefs approach the ground truth, which visually appears as the squares converging towards the color of the dot within them. When they coincide, the dot vanishes. An erroneous belief stands out as a red dot on blue square or, vice-versa, a blue dot on red square. Because we pre-sort the nodes in BnLearn datasets so that they are in topological order, the model must learn a lower triangle. This corresponds to a completely blue uppper triangle. Anti-causal violations are easily recognizable as non-blue squares in the upper triangle.

**Baseline comparisons** We compared our method to ICP (Peters et al., 2016) and Eaton & Murphy (2007a). Eaton & Murphy (2007a) handles uncertain interventions and Peters et al. (2016) handles unknown interventions. However, neither attempt to predict the intervention.

Table 1: **Baseline comparisons:** Cross entropy (lower is better) for edge probability on learned and ground-truth edges on Asia graph. compared to to Peters et al. (2016), (Eaton & Murphy, 2007a) and (Zheng et al., 2018)

| Our method | (Eaton & Murphy, 2007a) | (Peters et al., 2016) | (Zheng et al., 2018) |
|:---:|:---:|:---:|:---:|
| 0.0 | 0.0 | 10.7 | 3.1 |

**Importance of Dropout** To perform initial pretraining for an observational distribution, sampling adjacency matrices is required. One may be tempted to make these "fully-connected" (all-ones except for a zero diagonal), to give the MLP maximum freedom to learn any potential causal relations itself. We demonstrate that pretraining cannot be carried out this way, and that it is necessary to "drop out" each edge (with probability 0.5 in our experiments) during pre-training of the conditional distributions of the SCM. We attempt to recover the previously-recoverable graphs `chain3`, `fork3` and `confounder3` without dropout, but fail to do so, as shown in Figure 5.

**Generalization to Previously Unseen Interventions** It is often argued that learning approaches based on prediction do not necessarily yield models that generalize to unseen experiments, since they do not explicitly model changes through interventions - in contrast causal models use the concept

Table 2: **Evaluating the consequences of a previously unseen intervention:** (test log-likelihood under intervention)

|  | `fork3` | `chain3` | `confounder3` | `collider3` |
|---|---|---|---|---|
| **Our Model** | -0.4502 | -0.3801 | -0.2819 | -0.4677 |
| **Baseline** | -0.5036 | -0.4562 | -0.3628 | -0.5082 |

of interventions to explicitly model changing environments and hold thus the promise to work even under distributional shifts (Pearl, 2009; Schölkopf et al., 2012; Peters et al., 2017).

To test the robustness of causal modelling to previously unseen interventions (new values for an intervened variable), we evaluate a well-trained causal model against a non-causal variant model where all $c_{ij} = 1$, $i \neq j$. In both cases, an intervention is performed, and the models, with knowledge of the intervention, are asked to predict the rest. For this purpose, a batch of samples $X$ are drawn from $D_{\text{int}}$ and their average log-likelihoods are computed and contrasted. The intervention variable's contribution to the log-likelihood is ignored.

For all 3-variable graphs (`chain3`, `fork3`, `collider3`, `confounder3`), the causal model attributes higher log-likelihood to the intervention distribution's samples than the non-causal variant, thereby demonstrating causal models' superior generalization ability in transfer tasks. Table 2 collects these results.

**Importance of Inference** After the intervention has been performed, the learner draws data samples from the intervention distribution and computes the per-variable average log-probability under sampled adjacency matrices. The variable consistently producing the least-likely outputs is predicted to be the intervention node. Experiments over all 3-variable DAGs show that this prediction mechanism functions well in practice, yielding far above-random accuracy in correctly predicting the intervention node (Table 3), the model performance dropped significantly without the predication (Figure 6 Left) and is comparable to having the ground-truth intervention (Figure 6 Right).

Table 3: **Intervention Prediction Accuracy:** (identify on which variable the intervention took place)

| 3 variables | 4 variables | 5 variables | 8 variables |
|---|---|---|---|
| 95 % | 90 % | 81 % | 63 % |

**Effect of DAG Regularizer:** To promote the acyclicity of $C$, we include a DAG regularizer . This significantly improves cross-entropy of the solution (wrt. ground truth DAG) on all 3-variable graphs, as illustrated in Figure 6, and was therefore included in all experiments with $M > 3$ variables.

## 6 CONCLUSION

In this work, we introduced a framework for fast adaptation and slow learning of neural causal models. We demonstrate through experiments that the principle of optimizing an out-of-distribution meta-learning objective enables the learner to recover the causal graph structure for graphs with more than two variables. To achieve this we introduce an efficient parametrization of the belief regarding the underlying graph structure, implemented as an adaptive form of dropout on the inputs of MLPs computing the conditionals of the model. This relies on pre-training the conditionals using agnostic beliefs and by approximately inferring on which variable the intervention took place. We believe that our approach of treating seemingly observational data as being derived from an environment with agents executing interventions could represent an important change in modelling perspective with deeper implications.

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

# A APPENDIX

## A.1 DAG PENALTY DERIVATION

Recall, from Zheng et al. (2018):

**Theorem 1.** *A matrix* $\boldsymbol{W} \in \mathbb{R}^{d \times d}$ *is a DAG if and only if*

$$h(\boldsymbol{W}) = \operatorname{Tr}(e^{\boldsymbol{W} \circ \boldsymbol{W}}) - d = 0$$

For the two-variable ($d = 2$) graph with adjacency matrix

$$\boldsymbol{W} = \begin{pmatrix} 0 & \sigma(w_{12}) \\ \sigma(w_{21}) & 0 \end{pmatrix}$$

we have

$$\operatorname{Tr}(\exp(\boldsymbol{A})) = \operatorname{Tr} \sum_{n=0}^{\infty} \frac{\boldsymbol{A}^n}{n!}$$

$$\operatorname{Tr}(\exp(\boldsymbol{W} \circ \boldsymbol{W})) = \operatorname{Tr} \sum_{n=0}^{\infty} \frac{1}{n!} \begin{pmatrix} 0 & \sigma^2(w_{12}) \\ \sigma^2(w_{21}) & 0 \end{pmatrix}^n$$

There can only be even- or odd-length paths in a graph. Because, in a two-variable graph with no self-edges, all even-length paths are cycles and none of the odd-length paths are, we have

$$= \operatorname{Tr} \underbrace{\sum_{k=0}^{\infty} \frac{1}{(2k)!} \begin{pmatrix} 0 & \sigma^2(w_{12}) \\ \sigma^2(w_{21}) & 0 \end{pmatrix}^{2k}}_{\text{Even } n, \operatorname{Tr} \neq 0}$$

$$+ \operatorname{Tr} \underbrace{\sum_{k=0}^{\infty} \frac{1}{(2k)!} \begin{pmatrix} 0 & \sigma^2(w_{12}) \\ \sigma^2(w_{21}) & 0 \end{pmatrix}^{2k+1}}_{\text{Odd } n, \operatorname{Tr} = 0}$$

$$= \operatorname{Tr} \sum_{k=0}^{\infty} \frac{1}{(2k)!} \begin{pmatrix} \sigma^2(w_{12})\sigma^2(w_{21}) & 0 \\ 0 & \sigma^2(w_{12})\sigma^2(w_{21}) \end{pmatrix}^k$$

$$= 2 \sum_{k=0}^{\infty} \frac{\sigma^{2k}(w_{12})\sigma^{2k}(w_{21})}{(2k)!}$$

$$= \cosh(\sigma(w_{12})\sigma(w_{21}))$$

A pairwise generalization to multinode graphs over all $i \neq j$ is:

$$J_{\text{DAG}} = \sum_{i \neq j} \cosh(\sigma(w_{ij})\sigma(w_{ji}))$$

## A.2 TRAINING ALGORITHM

In this section, we describe the training algorithm in detail.

---

**Algorithm 1** Training Algorithm

---

1: **procedure** TRAINING(Categorical Distribution $D$, with $M$ nodes and $N$ categories)
2:     **Let** $i$ an integer from 0 to $M - 1$
3:
4:     **for** $k_{\text{pretrain}}$ steps **do**                    ▷ Pretraining Loop
5:         $x \sim D$                    ▷ Sample data from $D$
6:         $c \sim \text{Ber}(\sigma(\gamma))$              ▷ Sample config from structure distribution
7:         $L = -\log P(x|c)$        ▷ Compute log-probability of data given config
8:         $\theta_{\text{slow}} \leftarrow \text{Adam}(\theta_{\text{slow}}, \nabla_\theta L)$               ▷ Update $\theta_{\text{slow}}$
9:     **for** $k_{\text{intervention}}$ steps **do**               ▷ Interventions Loop
10:         `I_N` $\leftarrow$ `randint(0,` $M-1$`)`
11:         $D_{\text{int}} := D$ with intervention on node `I_N`
12:         **if** predicting intervention **then**
13:             $L_i \leftarrow 0 \quad \forall i$
14:             **for** $k_{\text{predict}}$ steps **do**          ▷ Prediction Loop
15:                 $x \sim D_{\text{int}}$             ▷ Draw batch of data from $D$
16:                 $c \sim \text{Ber}(\sigma(\gamma))$       ▷ Draw config from structure distribution
17:                 $L_i \leftarrow L_i + -\log P_i(x|c_i; \theta_{\text{slow}}) \forall i$    ▷ Accumulate NLL for every node $i$ separately
18:             `I_N` $\leftarrow \text{argmax}(L_i)$
19:         `gammagrads, logregrets = [], []`
20:         **for** $k_{\text{episode}}$ steps **do**              ▷ Transfer Episode Adaptation Loop
21:             $x \sim D_{\text{int}}$
22:             `gammagrad, logregret = 0, 0`
23:             **for** $k_{\text{cfg}}$ steps **do**           ▷ Configurations Loop
24:                 $c \sim \text{Ber}(\sigma(\gamma))$
25:                 $L_i = -\log P_i(x|c_i; \theta_{\text{slow}}) \quad \forall i$
26:                 `gammagrad += ` $\sigma(\gamma) - c$        ▷ Accumulate $\gamma$ gradient
27:                 `logregret += ` $\sum_{i \neq \texttt{I\_N}} L_i$      ▷ Accumulate regret
28:             `gammagrads.append(gammagrad)`
29:             `logregrets.append(logregret)`
30:         $J \leftarrow \lambda_{\text{MaxEnt}} L_{\text{MaxEnt}}(\gamma) + \lambda_{\text{Sparse}} L_{\text{Sparse}}(\gamma) + \lambda_{\text{DAG}} L_{\text{DAG}}(\gamma)$      ▷ Regularizers
31:         $\nabla_\gamma \leftarrow \nabla_\gamma J + \sum_k \texttt{gammagrads}_{kij}\texttt{logregrets.softmax(0)}_{ki}$    ▷ $\gamma$ gradient estimator
32:         $\gamma \leftarrow \text{Adam}(\gamma, \nabla_\gamma)$                  ▷ Update $\gamma$

---

## A.3 PRELIMINARIES

**Interventions**    In a purely-observational setting, it is known that causal graphs can be distinguished only up to a Markov equivalence class. In order to identify the true causal graph intervention data is needed (Eberhardt et al., 2012). Several types of common *interventions* may be available (Eaton & Murphy, 2007a). These are: *No intervention:* only observational data is obtained from the ground truth causal model. *Hard/perfect:* the value of a single or several variables is fixed and then ancestral sampling is performed on the other variables. *Soft/imperfect:* the conditional distribution of the variable on which the intervention is performed is changed. *Uncertain:* the learner is not sure of which variable exactly the intervention affected directly. Here we make use of soft interventions for several reasons: First, they include hard interventions as a limiting case and hence are more general. Second, in many real-world scenarios, it is more difficult to perform a hard intervention compared to a soft one. We also deal with a special case of uncertain interventions, where the variable selected for intervention is random and unknown. We call these *unidentified* or *unknown* interventions.

**Causal sufficiency**    The inability to distinguish which causal graph, within a Markov equivalence class, is the correct one in the purely-observational setting is called the *identifiability problem*. In our setting, all variables are observed (there are no latent confounders) and all interventions are

random and independent. Hence, within our setting the true causal graph is always identifiable in principle (Eberhardt et al., 2012; Heinze-Deml et al., 2018). We consider here situations where a single variable is randomly selected and intervened upon with a soft or imprecise intervention, its identity is unknown and must be inferred.

## A.4 EXPERIMENTAL SETUP

For all datasets, the weight parameters for the learned model is initialized randomly. In order to not bias the structural parameters, all $\gamma$ is initialized to $0.5$ in the beginning of training.

## A.5 SYNTHETIC DATA

SCM with $n$ variables is modeled by $n$ feedforward neural networks (MLPs) as described in section 3.1. For simplicity, we assume use an acyclic causal graph such that we could easily sample from it. Hence, given any pair of random variables $A$ and $B$, either $A \to B$, $B \to A$ or $A$ and $B$ are independent.

The MLP representing the ground-truth SCM has its weights $\theta$ initialized use orthogonal initialization with gain $2.5$ and the biases are initialized using a uniform initialization between $-1.1$ and $1.1$, which was empirically found to yield "interesting" yet learnable random SCMs.

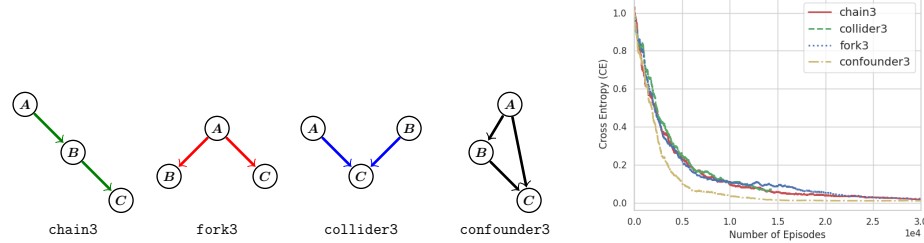

Figure 7: Left: Every possible 3-variable connected DAG. Right: Cross entropy for edge probability between learned and ground-truth SCM for all 3-variable SCMs.

## A.5.1 BNLEARN DATA REPOSITORY

The repo contains many datasets with various sizes and structures modeling different variables. We evaluate our model on 3 of the datasets in the repo, namely the Earthquake (Korb & Nicholson, 2010), Cancer (Korb & Nicholson, 2010) and Asia (Lauritzen & Spiegelhalter, 1988) datasets. The ground-truth model structure for the Cancer (Korb & Nicholson, 2010) and Earthquake (Korb & Nicholson, 2010) datasets are shown in Figure 8. Note that even though the structure for the 2 datasets seems to be the same, the conditional probability tables (CPTs) for these 2 datasets are very different and hence results in different structured causal models (SCMs) for the 2 datasets.

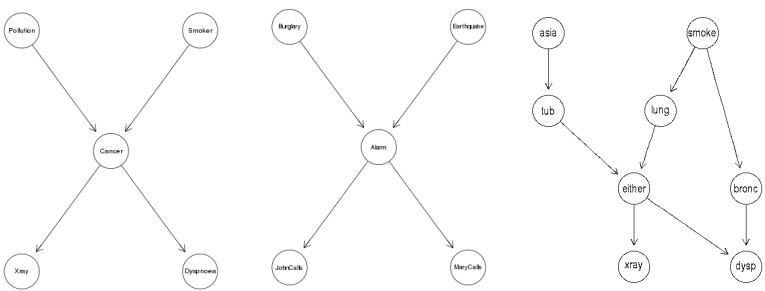

Figure 8: Left: Ground Truth SCM for Cancer. Middle: Groundtruth SCM for Earthquake. Right: Groundtruth SCM for Asia.

### A.5.2 TRAINING GROUND-TRUTH DATA GENERATOR

Because a CPT is capable of representing any distribution, and MLPs are strictly less powerful in this respect, it may not be possible to learn perfectly the distribution with our MLP learner model. We therefore train a near-ground-truth MLP to replicate as closely as possible the CPT's probability table, and then use this trained MLP are the ground-truth SCM data generator.

Training is by 1000 iterations of full-batch gradient descent with learning rate 0.001 and momentum 0.9, with all possible parent values masked with the ground-truth vector $\gamma_i$. The objective is to minimize mean squared error between the MLP's logits and the log-probability as drawn from the CPT. If the CPT contains a zero, it is approximated by a logit of $-100$.

Given that some of the CPTs contain very unlikely events, we have found it necessary to add a *temperature* parameter in order to make them more frequent. The near-ground-truth MLP model's logit outputs are divided by the temperature before being used for sampling. Temperatures above 1 result in more uniform distributions for all causal variables; Temperatures below 1 result in less uniform, sharper distributions that peak around the most likely value. We find empirically that a temperature of about 2 is required for our BnLearn benchmarks.

## A.6 EFFECT OF SPARSITY

We use a $L1$ regularizer on the structure parameters $\gamma$ to encourage a sparse representation of edges in the causal graph. In order to better understand the effect of the $L1$ regularizer, we conducted ablation studies on the $L1$ regularizer. It seems that the regularizer has an small effect on rate of converges and that the model converges faster with the regularizer. This is shown in Figure 9

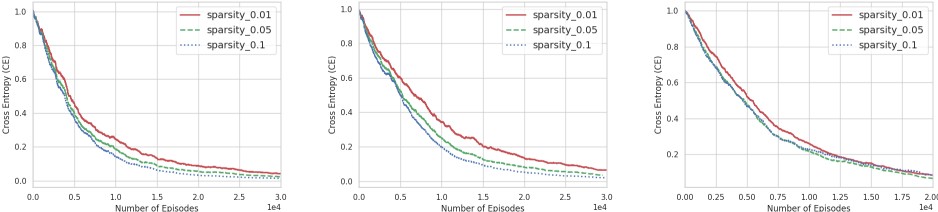

Figure 9: **Effect of Sparsity:** On 5 variable, 6 variable and 8 variable Nodes

## A.7 EFFECT OF TEMPERATURE

As noted in section A.5.1, we have introduced a temperature hyperparameter in order to encourage the groundtruth model to generate some very rare events in the conditional probability tables (CPTs) more frequently. We run ablation studies to understand the importance of the temperature term. A temperature of 1 corresponds to no changes to the underlying CPTs. As shown in Figure 12, for the Cancer (Korb & Nicholson, 2010) dataset, a temperature of 2 improves the accuracy of causal graph recovery.

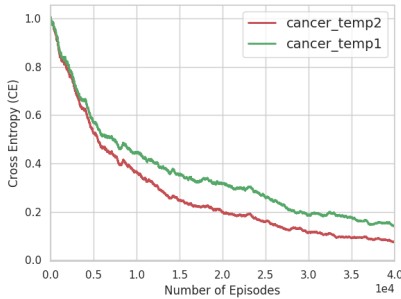

Figure 10: Cross entropy for edge probability between learned and ground-truth SCM for Cancer at varying temperatures.

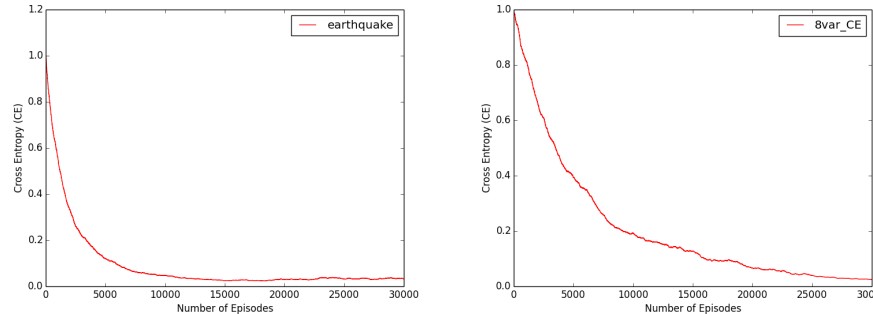

Figure 11: Cross entropy for edge probability between learned and ground-truth SCM. **Left**: The Earthquake dataset with 6 variables. **Right**: The Asia dataset with 8 variables

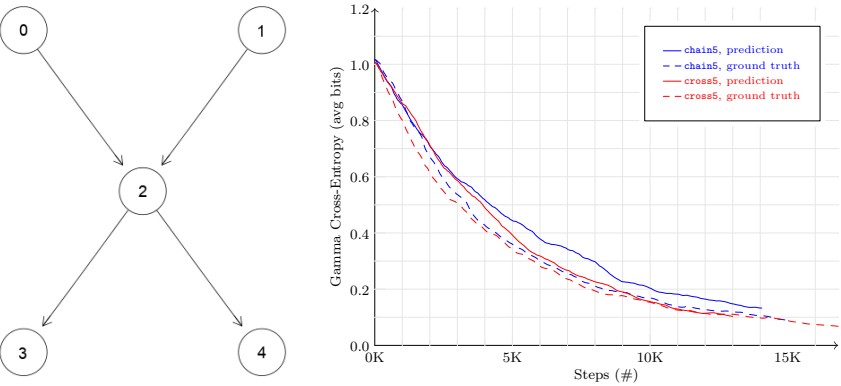

Figure 12: **Left:** SCM for `cross5` graph. **Right:** Cross entropy for edge probability between learned and ground-truth SCM for `chain5` and `cross5`, comparing the learning process with prediction of edges to ground-truth intervention

