# OpenReview forum: "Learning Neural Causal Models from Unknown Interventions"
_ICLR.cc/2020/Conference — Reject_

### Official Review · AnonReviewer2 · 2019-10-17
**Official Blind Review #2**

**Rating:** 8

**Review:**

The paper develops a learning-based causal inference method that performs multiple tasks jointly:

i) scalable discovery of the underlying Structured Causal Model (SCM) by modeling both structural assignments and the SCM as a continuously parameterized chain of distributions,

ii) identification of the intervened variables, which are not known to the model a-priori unlike the mainstream causal inference setups,

iii) achieving the two aforementioned goals using meta-learning driven heuristics, i.e. interventions cause distributional shifts.

While the paper adopts the core design choices from recent prior art (Bengio et al., 2019), the proposed methodology (especially ii)) is sufficiently novel to be published as a main-track conference paper. The paper is very well-written, follows a concrete and easy-to-follow story line. It solves multiple ambitious problems end-to-end and justifies the methodological novelty claims by a properly conducted set of experiments. The paper also successfully employs simple and useful but forgotten old techniques such as fast/slow parameter decomposition in the proposed model pipeline.

The intervention prediction heuristic is splendid. It is simple, sensible, and has been proven by experiments to be very effective. I would rate this as the primary novelty presented in this paper.

The paper can be improved if the below relatively minor concerns are addressed:

 i) It would be informative if the paper had a paragraph discussing also the fundamental limitations of the approach more openly. For instance, the choice of the neural net architecture used for the structural assignment might have a huge impact on the outcome, especially because the same architecture is repetitively used for all variables of the SCM. Furthermore, treatment of each variable with a fully independent neural net could cause overparameterization as the SCM grows in number of variables.

 ii) The paper makes a strong scalability claim across the variable size thanks to independent Bernoullis assigned on the adjacency matrix entries. However, it reports results only for very small SCMs. It is understandable that given the premature stage of the causal inference research might not grant standardized data sets at a larger scale, but at least lack of this quantitative scalability test could be acknowledged and the related claims could be a little bit softened.

 iii) I do not buy the argument in the first paragraph of Sec 3.5 about why the structural assignment functions need to be independent. As the model does not pose a distribution on neural net weights, sharing some weights (i.e. conditioning on them) would only bring conditional independence across the variables. I do not see a solid reason to try to avoid this. What is wrong for multiple variables to share some functional characteristics in their structural assignment? After all, some sort of conditional independence will be inevitable in modeling. If the variables share the same architecture, this is also conditional independence, not full independence. Relaxing the independence assumption and allowing some weight sharing could be beneficial at least for scalability of the model, could even bring about improved model fit due to cross-variable knowledge transfer.

Overall, none of the aforementioned three weaknesses is fundamental. In the status-quo, this is a spectactular research paper and my initial vote is an accept.

**Experience Assessment:**

I have read many papers in this area.

**Review Assessment: Checking Correctness Of Derivations And Theory:**

I carefully checked the derivations and theory.

**Review Assessment: Checking Correctness Of Experiments:**

I carefully checked the experiments.

**Review Assessment: Thoroughness In Paper Reading:**

I read the paper thoroughly.

---

> ### Author Response · Authors · 2019-11-14
> **Response to "Official Blind Review #2"**
>
> We are grateful to the reviewer for their enthusiastic feedback and comments!
>
> Q: “It would be informative if the paper had a paragraph discussing also the fundamental limitations of the approach more openly. For instance, the choice of the neural net architecture used for the structural assignment might have a huge impact on the outcome, especially because the same architecture is repetitively used for all variables of the SCM.”
>
> We thank the reviewer for pointing this out. We had chosen to implement all variables using the same neural network architecture for computational reasons (vectorization, batching), but it indeed might have had a significant impact on the learning process. A wider variety of architectures, incorporating heterogeneity in each variable’s model, would strengthen the case for the approach.
>
> There are, however, recent demonstrations that overparameterized neural networks can generalize well (with some regularization) [1]. This suggests that we may get away with deliberate over-parametrization, whether of each module separately or the whole network globally. The reviewer’s proposal below to allow some cross-variable parameter sharing is compatible with the latter; The right capacity and level of sharing for each variable would then be allocated according to the different pressures from the data and the training objective.
>
> [1]. Belkin, Mikhail, Daniel Hsu, Siyuan Ma, and Soumik Mandal. "Reconciling modern machine-learning practice and the classical bias–variance trade-off." Proceedings of the National Academy of Sciences 116, no. 32 (2019): 15849-15854.
>
> Q. “The paper makes a strong scalability claim across the variable size thanks to independent Bernoullis assigned on the adjacency matrix entries. However, it reports results only for very small SCMs. It is understandable that given the premature stage of the causal inference research might not grant standardized data sets at a larger scale, but at least lack of this quantitative scalability test could be acknowledged and the related claims could be a little bit softened.”
>
> We thank the reviewer for pointing this out. We agree with the reviewer’s point and a necessary continuation of our work is to demonstrate scaling to larger graphs available from e.g. the Bayesian Networks Repository. We will soften our scalability claims to better accord with the size of the problems solved in the paper.
>
> Q. “I do not buy the argument in the first paragraph of Sec 3.5 about why the structural assignment functions need to be independent. As the model does not pose a distribution on neural net weights, sharing some weights (i.e. conditioning on them) would only bring conditional independence across the variables. I do not see a solid reason to try to avoid this. What is wrong for multiple variables to share some functional characteristics in their structural assignment? After all, some sort of conditional independence will be inevitable in modeling. If the variables share the same architecture, this is also conditional independence, not full independence. Relaxing the independence assumption and allowing some weight sharing could be beneficial at least for scalability of the model, could even bring about improved model fit due to cross-variable knowledge transfer.”
>
> We appreciate the thought-provoking idea from the reviewer of cross-variable knowledge transfer via sharing. This is especially applicable to the real world setting, where it is likely that functional characteristics will be shared between variables of a similar nature. We will gladly investigate further this issue.
>
> Overall, we would like to thank the reviewer for the positive feedback and comments. We will perform the changes the reviewer recommends and relax the enforced independences to see if scalability or performance gains materialize.

---

### Official Review · AnonReviewer1 · 2019-10-19
**Official Blind Review #1**

**Rating:** 3

**Review:**

This paper proposes an SCM-model based on masked neural networks to capture arbitrary conditional relationships combined with meta-learning-style adaptation to reflect the effects of various unknown interventions. Overall the paper is well written and easy to follow, but some conceptual issues remain.


- How come there is hardly any discussion of the identifiability issue beyond the few sentences in A.3. This is one of the key issues in learning SCMs and it is strange that the concept of "faithfulness" is not even mentioned in the paper.

In general, there is hardly any discussion of what conditions are required for the proposed estimates to even be valid. The authors seem to be optimistically assuming that their neural network + metalearning model will somehow pick up on the correct structure, without any actual conceptual investigation of this issue.

- The massive downside of neural nets is all the various hyperparameters one has to set (eg. architecture, optimizer, activations, etc). In this setting, how do the authors propose selecting hyperparameter values? How does the reader know the authors did not simply tune their hyperparameters to best match the underlying ground truth (I assume the proposed methodology has many more hyperparameters and thus more degrees of freedom here).
I would like to see the empirical performance of different variants of your model with different hyperparameter values to assess its sensitivity to these choices.

- Why does one even care about the graph being acylic in this setting?
The mere fact that the authors require a regularizer to ensure acylicity suggests this approach is prone to mis-identifying the ground truth structure (which is always acyclic in the experiments).

- One main reason for SCM modeling in science and policy-making is for analysts to better understand the data generating phenomena.  However your use of neural networks here seems to hamper interpretability, so how do you reconcile this issue? Also is your sparsity regularizer satisfactory to confidently diagnose presence/absence of an edge (in constrast to statistical hypothesis tests, say based on conditional independence). Isn't this heavily influenced by the particular sparsity-regularizer value that happened to be selected?


- Related papers that utilize the same idea of predicting a variable conditioned on subset of other variables via neural network + masking strategy:

Ivanov et al (2019). VARIATIONAL AUTOENCODER WITH ARBITRARY CONDITIONING.
https://openreview.net/pdf?id=SyxtJh0qYm

Li et al (2019). Flow Models for Arbitrary Conditional Likelihoods.
https://arxiv.org/abs/1909.06319

Yoon et al. GAIN: Missing data imputation using generative adversarial nets. Proceedings of the 35th International Conference on Machine Learning, volume 80 of Proceedings of Machine Learning Research, 2018. http://proceedings.mlr.press/v80/yoon18a.html

Douglas et al. A universal marginalizer for amortized inference in generative
models. arXiv preprint arXiv:1711.00695, 2017

For clarity, the authors should highlight the differences of their approach from these works (beyond the causal setting).

- Given the lack of theoretical / conceptual guarantees that the methodology will work, our faith in the proposed methodology rests entirely on the empirical experiments.  However, I find these a bit too basic to be very convincing, and would at least like to see more methods being compared (in particular for the simulated graphs as well).

- The authors should describe what are the underlying interventions in each dataset a bit more.

- The Figures should be better explained (took me a while to figure out what dots/colors represent).

- Why do the authors report cross entropy loss in Table 1? To my knowledge this is not a standard metric for measuring the quality of structure-estimates.

- Instead of ICP (which is constrained to be linear which is unrealistically simple), why don't the authors compare against nonlinearICP (which is more flexible like their neural networks):

Heinze-Deml et al (2018). Invariant Causal Prediction for Nonlinear Models. https://arxiv.org/pdf/1706.08576.pdf

**Experience Assessment:**

I have published one or two papers in this area.

**Review Assessment: Checking Correctness Of Derivations And Theory:**

N/A

**Review Assessment: Checking Correctness Of Experiments:**

I carefully checked the experiments.

**Review Assessment: Thoroughness In Paper Reading:**

I read the paper thoroughly.

---

> ### Author Response · Authors · 2019-11-12
> **Response to Official Blind Review #1 (Part 1)**
>
> We thank the reviewer for such detailed feedback. We are conducting additional experiments based on the feedback and will update the paper and rebuttal once the experiments are completed.
>
> The reviewer expresses several general concerns about the use of neural networks for causal inference, focusing on attributes such as their large design space and their interpretability. We would like to underscore that this paper is intended as a step from today’s completely non-causal neural networks towards incorporating more of the abilities required for handling causality. As such, our proposed method will indeed retain most of the benefits and limitations of neural networks, but improve on them by identifying causal structures.
>
> Q. ”How come there is hardly any discussion of the identifiability issue beyond the few sentences in A.3. This is one of the key issues in learning SCMs and it is strange that the concept of "faithfulness" is not even mentioned in the paper. ….In general, there is hardly any discussion of what conditions are required for the proposed estimates to even be valid. The authors seem to be optimistically assuming that their neural network + metalearning model will somehow pick up on the correct structure, without any actual conceptual investigation of this issue.”
>
> Because our task setup allows a random intervention over any variable, per (Eberhardt et al., 2012) it is at least in theory possible to identify the correct graph. The rest of the paper was mostly directed at showing that this is not only possible in theory but in practice as well.
>
> We thank the reviewer for mentioning that some discussion of faithfulness would enhance the paper. There are several aspects that are relevant as listed below.
> Our model does indeed assume faithfulness, however, this is not a limitation in practice. Because of the continuous evolution of the functional parameters for the conditional distributions MLP, we believe that occurrences of unfaithful populations will be extremely short-lived and exceedingly rare to begin with. Lastly, because our procedure invokes an outer-loop optimization procedure, gradient estimate errors induced by unfaithfulness can be compensated.
> The faithfulness assumption (Pearl 2009, Peters et al. 2017) implies that any d-separation in the graph corresponds to a conditional independence in the data generating random variables. Under the assumption of faithfulness and a sufficiently large sample size, the Markov blanket can consistently be recovered given the availability of an efficient feature selection algorithm [1]. Neural Networks have been shown to be able to learn good features [2, 3] and require large datasets for training, which we assumed to be given here. We agree with the reviewer that similarly to the already mentioned assumptions of Markov equivalence and causal sufficiency (see A.3 PRELIMINARIES) we will add a discussion on faithfulness and the assumptions of the availability of large datasets to the manuscript.
> [1] J.-P. Pellet and A. Elisseeff. Using markov blankets for causal structure learning. Journal of Machine Learning Research, 2008
> [2] Bengio, Yoshua. Learning deep architectures for AI. Foundations and trends in Machine Learning, 2009
> [3] Bengio, Yoshua et. al, Representation learning: A review and new perspectives, arxiv 1206.5538, 2012

---

> ### Author Response · Authors · 2019-11-12
> **Response to Official Blind Review #1 (Part 2)**
>
> Q. ”In this setting, how do the authors propose selecting hyperparameter values? How does the reader know the authors did not simply tune their hyperparameters to best match the underlying ground truth (I assume the proposed methodology has many more hyperparameters and thus more degrees of freedom here).”
>
> Very little effort was required for tuning the neural network hyperparameters:
>
> 1. All of our experiments (synthetic and real data) use the same hyperparameters unless otherwise specified.
> 2. We used common strategies for training a neural network and this does usually include several hyperparameters. Among others, there are the specific architecture, activation, number of hidden layers, size of hidden layers, learning rate and optimizer.
>      a.  The choice of the architecture was the simplest feedforward neural network, an MLP, with
>      b.  The smallest possible number of hidden layers which is 1
>      c.  The number of hidden neurons was designed only to be greater than the number of input or output neurons.
>      d. Given that ReLUs are standard in the literature, we selected a simple, well-known variant called LeakyReLU that avoids a common problem called the dying neuron problem [1].
>           i. The alpha parameter was arbitrarily set to 0.1 and never tuned.
>           ii. Since we are training (a set of) MLPs, we adapted some of the commonly used strategies for training MLPs. We used the Adam optimizer, one of the most successful ones in the literature [2] and selected the best learning rate from [0.01, 0.05, 0.001, 0.005].
>   3.  We are running additional experiments with various size of hidden units. We will update our paper with the new results once these experiments are completed.
>   4. For reproducibility, future benchmarking and baseline comparisons, all code will be released.
>
>    [1]. Lu, Lu, Yeonjong Shin, Yanhui Su, and George Em Karniadakis. "Dying ReLU and Initialization: Theory and Numerical Examples." arXiv preprint arXiv:1903.06733 (2019).
>
>    [2]. Kingma, Diederik P., and Jimmy Ba. "Adam: A method for stochastic optimization." arXiv preprint arXiv:1412.6980 (2014).
>
> Q. “Why does one even care about the graph being acyclic in this setting?”
>
> We defined our groundtruth SCM to be acyclic for the simplicity of sampling, otherwise we could not perform ancestral sampling. That being the case, an acyclic regularizer restricts the set of solutions, encouraging faster convergence of the model from a statistical point of view. Adding the regularizer speeds up convergence, but asymptotically both models with and without regularization converge towards the same point.
>
> Q. “One main reason for SCM modeling in science and policy-making is for analysts to better understand the data generating phenomena.  However your use of neural networks here seems to hamper interpretability, so how do you reconcile this issue?”
>
> In the foreword, we had lightly touched on the general concerns raised about neural networks and their interpretability. We will dive into greater detail here.
>
> The (learned) structural parameters, which define the causal structure of the solution, are directly interpretable as an adjacency matrix. Examples of the learned adjacency matrix extracted from our model can be found in Figure 3 and 4 in the paper.
>
> As regards the MLP-parametrized conditionals, they are as interpretable as conditional probability tables. This is because the MLP’s learned functional parameters can always be reduced to such a table by querying the MLP for all possible discrete values of all possible ancestors.
>
> There is a vast literature on interpretability of deep learning, of course, but we must admit that our main goal is to design better learning algorithms for autonomous intelligent systems (like robots) where the ability of those systems to understand the world is the primary goal (as opposed to extracting that knowledge for human consumption). Our neural networks’ interpretability by analysts was therefore only a secondary objective, although in the end the model remains quite interpretable.

---

> ### Author Response · Authors · 2019-11-12
> **Response to Official Blind Review #1 (Part 3)**
>
> Q. “Why does one even care about the graph being acyclic in this setting?”
>
> We defined our groundtruth SCM to be acyclic for the simplicity of sampling, otherwise we could not perform ancestral sampling. That being the case, an acyclic regularizer restricts the set of solutions, encouraging faster convergence of the model from a statistical point of view. Adding the regularizer speeds up convergence, but asymptotically both models with and without regularization converge towards the same point.
>
> Q. “One main reason for SCM modeling in science and policy-making is for analysts to better understand the data generating phenomena.  However your use of neural networks here seems to hamper interpretability, so how do you reconcile this issue?”
>
> In the foreword, we had lightly touched on the general concerns raised about neural networks and their interpretability. We will dive into greater detail here.
>
> The (learned) structural parameters, which define the causal structure of the solution, are directly interpretable as an adjacency matrix. Examples of the learned adjacency matrix extracted from our model can be found in Figure 3 and 4 in the paper.
>
> As regards the MLP-parametrized conditionals, they are as interpretable as conditional probability tables. This is because the MLP’s learned functional parameters can always be reduced to such a table by querying the MLP for all possible discrete values of all possible ancestors.
>
> There is a vast literature on interpretability of deep learning, of course, but we must admit that our main goal is to design better learning algorithms for autonomous intelligent systems (like robots) where the ability of those systems to understand the world is the primary goal (as opposed to extracting that knowledge for human consumption). Our neural networks’ interpretability by analysts was therefore only a secondary objective, although in the end the model remains quite interpretable.
>
> Q. “Related papers that utilize the same idea of predicting a variable conditioned on subset of other variables via neural network + masking strategy:”
>
> We thank the reviewers for pointing such a complete list of papers, they are indeed relevant and we will update our relevant work section with the list of papers.
>
> Q “our faith in the proposed methodology rests entirely on the empirical experiments.  However, I find these a bit too basic to be very convincing, and would at least like to see more methods being compared (in particular for the simulated graphs as well).”
>
> We thank the reviewers for pointing this out. There are several aspects that are relevant as listed below.
>
>  a. As mentioned in other recent works e.g. [1], ICP is one of the state-of-the-art algorithms. We would refer to the extensive experiments in their paper for additional baselines comparisons and while we can likewise add more baselines, we do not expect any changes in results. Please likewise see our answer for the comparison against non-linear ICP below.
>
>  [1]. Versteg, Boosting Local Causal Discovery in High-Dimensional Expression Data https://arxiv.org/pdf/1910.02505v2.pdf
>
>   b. We have examined an array of causal learning methods where an open-source implementation is available. However, few of these are applicable to discrete data from interventions. Many of these methods can only handle continuous data (not discrete data) e.g. LinGAM and many others do not handle interventions. Hence we only compared to the ones that we had in the paper. If the reviewer is aware of an implementation of an algorithm applicable to our setup (leaving aside non-linear ICP, which we discuss in the answer below), we would be more than happy to run it.
>
>    c. We also present experiments aimed at measuring generalization (in terms of predictive power and likelihood) using the learned causal structure. Ideally, If the model has learned the right structure, it should generalize better. This is shown in Table 3 of our paper.
>
>
> Q. “The authors should describe what are the underlying interventions in each dataset a bit more.”
>
> We thank the reviewer for pointing this out, this is a good point and we will include this in the next revision.

---

> ### Author Response · Authors · 2019-11-12
> **Response to Official Blind Review #1 (Part 4)**
>
> Q. “Why do the authors report cross entropy loss in Table 1?”
>
> We appreciate the reviewer for pointing this out. We reported cross entropy because we learn the likelihood of edges between variables rather than iterating through all possible graphs (which is typically done e.g. in ICP). Hence we maintain a distribution over graphs and we need to score how good that distribution is. This loss thus gives a better indication of how our method learned and converged over time.  There is also a direct comparison to the ground-truth graph and the cross entropy should converge close to 0 if our model has learned the correct structure. On top of reporting cross entropy, we also evaluate our model on predicted likelihood for out-of-distribution generalization as shown in Table 3.
>
> Q. “Instead of ICP (which is constrained to be linear which is unrealistically simple), why don't the authors compare against nonlinearICP (which is more flexible like their neural networks): “
>
> While we agree that the non-linearity in the non-linear version of ICP adds flexibility, it likewise increases the difficulty since it is unclear which non-linear and non-parameteric conditional independence test to use in practice. The performance of nonlinear ICP critically depends on the conditional independence tests. That is one reason why in the non-linear ICP paper it is explicitly recommended for practical purposes to use non-linear ICP (over its linear version, see discussion p24-p25 in (1)) only if all linear models are rejected. However, not all linear models were rejected by ICP in our case. Moreover conditional independence testing was shown to be hard [1] which might be one reason why our method shows superior performance over the state-of-the-art method.
>
> [1]. Shah, Rajen D., and Jonas Peters. "The hardness of conditional independence testing and the generalised covariance measure." arXiv preprint arXiv:1804.07203 (2018).

---

> ### Author Response · Authors · 2019-11-14
> **Revised paper uploaded**
>
> Dear reviewer #1,
>
> We’d like to thank you again for your review and feedback! We have updated our paper with your suggestions and those of others. In particular, we made the suggested citations to related work in section 4, included a section explaining the intervention in section 3.5, and began re-running the experiments with 5 random seeds each and reporting the error bars (Figure 5 Left is a beginning). We are also running experiments while varying the number of hidden states, as you have suggested. Would you have any other questions regarding the rebuttal?  We would be happy to provide further revisions or experiments to address any remaining issues. Many thanks again for your review and feedback.

---

> > ### Comment · AnonReviewer1 · 2019-11-15
> > **Comments re revision**
> >
> > I applaud the authors for making good revisions to their paper.
> >
> > However, my main concern still stands.  Without any theoretical/conceptual underpinnings of when the proposed methodology will/won't work, all support for the proposed methodology must come from the empirical experiments.  Unfortunately, I still do not believe experiments are comprehensive enough to convince me of the strength of the proposed methodology.  In particular, there are simply insufficient comparisons against other methods.  The proposed methodology does not appear better than Eaton & Murphy (2007), and the ICP method is clearly inappropriate as a linear model being applied in a setting where the underlying (simulated) relationships are known to be nonlinear (in fact match the MLP used in the authors method which is even more unfair).  Finally, the method of Zheng et al (2018) depends on various hyperparameters, and it is unclear whether these were set more or less favorably than the authors' method.

---

> > > ### Author Response · Authors · 2019-11-15
> > > **Response to Reviewer 1**
> > >
> > > We thank the reviewer for his feedback. We would like to point out that:
> > >
> > > (a) We agree that the concern about hyper-parameter selection is valid for all neural network based methods but would like to point out that for each method we applied the same budget for hyper-parameter search. In addition, for full clarity and future benchmarking all our code will be released and made accessible.
> > >
> > > (b) We politely disagree with the reviewer that the comparison is inappropriate or unfair. The asia graph (which is defined in the BN Learn repository) dataset we use to evaluate all comparison methods is a real-world dataset and the underlying relationships are not known. In particular, they were not simulated from an MLP.
> > >
> > > (c) While we agree that the non-linearity in non-linear ICP adds flexible, the method has the fundamental problem that it relies on conditional independence testing, which is hard (Peters and Shah 2019). As pointed out by the author’s of non-linear ICP themselves and referenced in our response Part 3, a comparison is not recommended “In practice” (Conclusion p.24 in non-linear ICP, Heinze-Deml et.al 2018). Nevertheless, we will try to add a comparison to non-linear ICP.
> > >
> > > (d) We would like to point out that we already compare against 3 methods, in particular to the state-of-the art method for causal induction from interventional data, as noted in [1]. In general, we examined an array of causal learning methods where an open-source implementation is available. However, many of these methods can only handle continuous data (not discrete data) e.g. LinGAM, while many others do not handle interventions. We compare against all methods, which are applicable in our case, provide open-source code and were the authors themselves do not provide alternative recommendations on how to proceed in practice.
> > >
> > > [1] Versteg, Boosting Local Causal Discovery in High-Dimensional Expression Data https://arxiv.org/pdf/1910.02505v2.pdf

---

> > > > ### Author Response · Authors · 2019-11-15
> > > > **Rebuttal Discussion**
> > > >
> > > > Dear Reviewer,
> > > >
> > > > Could you let us know if our response has addressed the concerns raised in your review? I think our response in point (b) above clarifies your main concern about insufficient comparisons (as in Asia graph, it was not simulated from MLP).
> > > >
> > > > We would be happy to provide further revisions  to address any remaining issues and would appreciate a response from you on the points that we raised (as rebuttal period is going to end soonish).
> > > >
> > > > Thanks for taking time and discussing with the authors. We appreciate it. :)

---

### Official Review · AnonReviewer3 · 2019-10-23
**Official Blind Review #3**

**Rating:** 6

**Review:**

This paper proposes a MAML objective to learn causal graphs from data. The data in question is randomized but the algorithm does not have access to the identity of the intervention variable. So there is an added layer of complexity of deciphering which variable was intervened on. The MAML objective, in this case, links the causal structure to the slow-moving parameter theta_slow.

The novelty of the paper seems to be in the application of the MAML framework to causal discovery which is interesting to me. I think a little theory about the sensitivity of the claim of ' theta slow changes relate to the causal structure ' is important. Even showing empirically which sort of graphs and functions become issues for the model would be useful.

Here are my issues with the paper:
 - No error bars for cross-entropy are reported in the experiments.
 - The acyclic regularizer does not reject large length cycles than 3.
 - The ability to predict interventions seems to drop off sharply as the number of nodes increases. This suggests an inability to scale to more than 20 variables.
 - The experimental setup of uniformly sampling an intervening variable seems artificial to me.
 - MLP-specification of the SCM also seemed a bit artificial to me.

Overall, the experiments look reasonable and the method itself seems interesting although further work is needed to show it is useful.

(writing comments) The paper could use a more structured re-write. I had trouble tracking terms around the paper. For example, there seems to be a difference between P_i and P because the former uses theta_i and the latter only uses theta_slow only.

---------------------------------

Updated score to 6 after rebuttal.

**Experience Assessment:**

I have read many papers in this area.

**Review Assessment: Checking Correctness Of Derivations And Theory:**

I assessed the sensibility of the derivations and theory.

**Review Assessment: Checking Correctness Of Experiments:**

I assessed the sensibility of the experiments.

**Review Assessment: Thoroughness In Paper Reading:**

I read the paper at least twice and used my best judgement in assessing the paper.

---

> ### Author Response · Authors · 2019-11-12
> **Response to Official Blind Review #3 (Part 1)**
>
> We thank the reviewer for the feedback. We have conducted additional experiments based on the feedback and will update our paper once the experiments are completed.
>
> Our paper is related to MAML like procedures for meta-learning, but goes beyond the usual setting, making a significant contribution through developing more sophisticated algorithms that enable causal structure learning.
> The difficulties those changes addressed are intrinsic to causal structure learning, especially in the challenging unknown-intervention scenario that we have set ourselves. The challenges we solve are 1) how to handle unknown interventions, 2) how to avoid an exponential search over all possible DAGs, 3) how to model the effect of the intervention, and finally 4) how to model the underlying causal structure.
>
> Q. “No error bars for cross-entropy are reported in the experiments.”
> We thank the reviewer for pointing this out. We have conducted additional experiments and will update our paper once the experiments have been completed.
>
> Q. “The acyclic regularizer does not reject large length cycles than 3.‘
> We appreciate the reviewer’s concern. The regularizer can be extended to length-n cycles, however, this becomes more computationally demanding as n increases. However, we found that a smaller n does not affect our model empirically.  As shown in Figure 2, 3 and 4, our model did not learn cycles of any length greater than 2. In fact, we have found that even completely removing this regularizer does not hurt the asymptotic performance of our model. The regularizer helps the model to converge faster, however, the model still converges reasonably fast without the regularizer, as shown in Figure 6 Right.
>
> Q. “The ability to predict interventions seems to drop off sharply as the number of nodes increases.”
> We are aware of this limitation. It makes sense that  guessing which node has been intervened becomes harder as the number of nodes increases and we find that empirically, without surprise. We agree  that it is a challenge to scale to larger graphs (namely graphs with more than 20 variables), however even for the sizes of graphs we consider our paper finds greatly improved solutions and this is already a significant advance over past work.  One extension we hope will help to overcome this difficulty would be to perform a soft prediction of the interventional nodes, instead of the hard decision that we have now. We also would like to highlight that the intervention prediction performs significantly better than random at all times.
>
> One note on the recent papers on this topic: although ICP and non-linear ICP consider a larger number of covariates, they only aim to identify the causal parents of one variable. This task alone already has exponential cost, which would be further increased if the algorithm were applied for reconstructing the whole graph by applying it  iteratively to each node. Due to the computational cost, this is infeasible for larger graphs.. Other recent papers e.g. [1] likewise only consider similar number of variables given the computational cost of the proposed algorithms. In contrast one contribution of our paper is a proposal how to avoid an exponential search over all possible DAGs.
>
> [1]. Ghassami, AmirEmad, Saber Salehkaleybar, Negar Kiyavash, and Kun Zhang. "Learning causal structures using regression invariance." In Advances in Neural Information Processing Systems, pp. 3011-3021. 2017.
>
> Q. “The experimental setup of uniformly sampling an intervening variable seems artificial to me.”
>
> We thank the reviewer for pointing this out. We agree that in the real world, interventions rarely appear to be chosen uniformly randomly. However, given the lack of better real-world causal structures than those from the BNLearn graph repository, and the lack of a commonly-agreed intervention probability on each node, uniform sampling seemed reasonable. Doing otherwise would have required us to justify why we picked those specific intervention probabilities. However, if the reviewer has suggestions for specific non-uniform intervention probabilities, we will be happy to perform additional experiments with them.

---

> > ### Comment · AnonReviewer3 · 2019-11-13
> > **Opinion changed a little**
> >
> > Regarding contributions:
> > I agree that these contributions are noteworthy.
> >
> > However, the idea of masking was used in https://arxiv.org/pdf/1803.04929.pdf which your method also depends on. I believe that this same technique gives you both the ability to model the causal structure and avoids exponential search, correct?
> >
> > Could you also clarify why the comparison against this work was not done? Unless I am missing something, while Kalainathan et al. learn from observational data, the method can be run on your setup. And they do not suffer from the exponential time-complexity.
> >
> > Error bars:
> > Thank you for this. I am interested in seeing how well MAML does.
> >
> > About cyclic regularizer:
> > Thank you for pointing this out.
> >
> > About predicting interventions:
> > You claim in the paper that "We find that ignoring this issue considerably hurts or slows down meta-learning, suggesting that we should try to infer on which variable the intervention took place."
> > So this seems strongly coupled with the ability of MAML to recover causal structure. Then, am I correct in saying that MAML's quality of causal discovery is maintained even when the intervention prediction quality goes down?
> >
> > Maybe I'm missing something but if the intervention cannot be predicted, does the setup not boil down to estimating structure from observational data?
> >
> > Uniform sampling of intervention:
> > I think it is more interesting to restrict the set of nodes you perform interventions on. This is sort of like a held-out intervention evaluation of your method. I think I was unclear about my issue here. It was not the uniformity, rather it was that the interventions were being done on all nodes.
> >
> > Overall, I believe this method shows promise but needs a little more evaluation and understanding.

---

> > > ### Author Response · Authors · 2019-11-14
> > > **Response to "Opinion changed a little"**
> > >
> > > We thank the reviewer for the very prompt response and we thank the reviewer for noting our contributions. We have begun updating our paper’s existing figures with error bars and are running the reviewer’s suggested additional experiments.
> > >
> > > Q. “Error bars: Thank you for this. I am interested in seeing how well MAML does.”
> > >
> > > A. We uploaded a new revision of the paper and updated Figure 5 Left in the paper to reflect the errors bars. We have conducted experiments for all 3-variable graphs with PRNG seeds 1 to 5. The graphs for remaining experiments will be updated in due course, but limited compute resources may delay them beyond the rebuttals deadline.
> > >
> > > Q. ”However, the idea of masking was used in https://arxiv.org/pdf/1803.04929.pdf which your method also depends on. I believe that this same technique gives you both the ability to model the causal structure and avoids exponential search, correct? Could you also clarify why the comparison against this work was not done? Unless I am missing something, while Kalainathan et al. learn from observational data, the method can be run on your setup. And they do not suffer from the exponential time-complexity.”
> > >
> > > A. We appreciate the reviewer for pointing us to this work. It is true that this technique would also have the ability to model the causal structure and avoids the exponential search.
> > >
> > > The main reason we did not compare to Kalainathan et al. was because their technique only handles continuous data and our method tackles discrete data. We settled on the discrete case because we needed datasets that are large and allow for interventions. We are aware of no large, multi-variable datasets for continuous variables, and an effort [1] to create such a dataset is only in its infancy: It supports only two variables (cause and effect pairs) and its authors themselves have made an urgent public call for far more validation pairs. By contrast, the Bayesian Networks Repository has a variety of discrete, multi-variable networks publicly available for benchmarking.
> > >
> > > An additional complication is Kalainathan et al.’s use of a GAN framework, which is not trivial to extend to the discrete case. The authors themselves admit as much in the conclusion of their paper: “An on-going extension regards the case of categorical and mixed variables, taking inspiration from discrete GANs (Hjelm et al., 2017).” As of today, no such extension has been published.
> > >
> > > We compared to 3 other methods. In particular, we compared to ICP, which is one of the state-of-the-art methods for causal induction from interventional data, as noted in [2]. In general, we examined an array of causal learning methods where an open-source implementation is available. However, many of these methods can only handle continuous data (not discrete data) e.g. LinGAM, while many others do not handle interventions.
> > >
> > > [1] J. M. Mooij, J. Peters, D. Janzing, J. Zscheischler, B. Schoelkopf: "Distinguishing cause from effect using observational data: methods and benchmarks", Journal of Machine Learning Research 17(32):1-102, 2016
> > > [2]. Versteg, Boosting Local Causal Discovery in High-Dimensional Expression Data https://arxiv.org/pdf/1910.02505v2.pdf
> > >
> > > Q. “Uniformly sampling of intervention:
> > > I think it is much more interesting to restrict the set of nodes you perform interventions on. This is sort of like a held-out intervention evaluation of your method. I think I was unclear about my issue here. It was not the uniformity, rather it was that the interventions were being done on all nodes.”
> > >
> > > A. We thank the reviewer for clarifying this point. Although many SCMs’ true causal structures can be recovered with only a restricted set of interventions, in the general case one needs the ability to intervene on all variables, and this is why we allow the method to do so. That being said, the reviewer’s proposed experiments would be a valuable addition to the paper, and although limited computing resources and a backlog of other suggested experiments might prevent us from inserting these before the close of the rebuttal period, we will dedicate a series of experiments to this topic.

---

> ### Author Response · Authors · 2019-11-12
> **Response to Official Blind Review #3 (Part 2)**
>
> Q. “MLP-specification of the SCM also seemed a bit artificial to me”.
>
> Both the groundtruth SCM and our model are parameterized by MLPs.
> Could we ask the reviewer to clarify if there is a specific aspect of this setup using an MLP the reviewer finds artificial here?  MLPs are used successfully in a large number of state-of-the-art solutions to many real-world ML problems. We think of our work as a step to bring causality to deep learning, which as Pearl would say, would be helpful to further climb the ladder of intelligence.
>
> We chose to parameterize the ground truth SCM by MLPs for the ease of defining the conditional probability table (CPT), such that we do not have to exhaustively define the CPT for different variables and graphs, something very convenient  as the number of variables increases (and the size of a full CPT would grow exponentially).
> As for our model, one of the key contributions is to parametrize a learned SCM using a neural network. It is suggested by the recent review paper on causal structural learning. The paper [1] concluded that "more efficient algorithms are needed". One possibility of a more efficient algorithm is one that avoids an explicit exponential search over all possible DAGs and our framework of learning a SCM parameterized by a neural network using a meta-learning approach is a step towards this goal.
>
> Another contribution is that our framework/ method of MLP specification of the SCM generalizes well to the challenge of out-of-distribution interventions.
>
> [1]. Heinze-Deml, Christina, Marloes H. Maathuis, and Nicolai Meinshausen. "Causal structure learning." Annual Review of Statistics and Its Application 5 (2018): 371-391.
>
> Q. “ had trouble tracking terms around the paper.”
>
> We thank the reviewer for pointing this out. We have double checked our use of terms and updated the paper to improve clarity in this regard.

---

### Decision · Program_Chairs · 2019-12-19

**Decision:**

Reject

**Comment:**

This paper proposes a metalearning objective to infer causal graphs from data based on masked neural networks to capture arbitrary conditional relationships. While the authors agree that the paper contains various interesting ideas, the theoretical and conceptual underpinnings of the proposed methodology are still lacking and the experiments cannot sufficiently make up for this. The method is definitely worth exploring more and a revision is likely to be accepted at another venue.